# DiffPoseTalk: Speech-Driven Stylistic 3D Facial Animation and Head Pose Generation via Diffusion Models

## Abstract

The generation of stylistic 3D facial animations driven by speech poses a significant challenge as it requires learning a many-to-many mapping between speech, style, and the corresponding natural facial motion. However, existing methods either employ a deterministic model for speech-to-motion mapping or encode the style using a one-hot encoding scheme. Notably, the one-hot encoding approach fails to capture the complexity of the style and thus limits generalization ability. In this paper, we propose DiffPoseTalk, a generative framework based on the diffusion model combined with a style encoder that extracts style embeddings from short reference videos. During inference, we employ classifier-free guidance to guide the generation process based on the speech and style. We extend this to include the generation of head poses, thereby enhancing user perception. Additionally, we address the shortage of scanned 3D talking face data by training our model on reconstructed 3DMM parameters from a high-quality, in-the-wild audio-visual dataset. Our extensive experiments and user study demonstrate that our approach outperforms state-of-the-art methods. The code and dataset will be made publicly available.

## 1 Introduction

The domain of speech-driven 3D facial animation has experienced significant growth in both academia and industry, primarily owing to its diverse applications in education, entertainment, and virtual reality. Speech-driven 3D facial animation generates lip-synchronized facial expressions from an arbitrary speech signal. It is a highly challenging research problem due to the cross-modal many-to-many mapping between the speech and the 3D facial animation. However, most existing speech-driven 3D facial animation methods rely on deterministic models (Cudeiro et al., 2019; Richard et al., 2021; Fan et al., 2022; Xing et al., 2023), which often fail to sufficiently capture the complex many-to-many relationships and suffer from the regression-to-mean problem, thereby resulting in over-smoothed face motions. Furthermore, these methods generally employ a one-hot encoding scheme for representing speaking styles during training, thus limiting their adaptability to new speaking styles.

As a typical probabilistic model, diffusion models can eliminate the over-smoothed face motions caused by most existing methods based on regression models. Compared with the deterministic models, diffusion models can fit various forms of distributions and thus better cope with the many-to-many mapping problem. Recent diffusion models have shown impressive results in various domains (Yang et al., 2022). Specifically, the existing diffusion-based audio-driven human motion generation methods have shown appealing results. However, they are not trivially transferable to speech-driven facial animation for three main reasons. First, unlike gestures or dance motions, which have a more relaxed temporal correlation with audio, facial movements — particularly lip motions — require precise alignment with speech. Second, the semantic richness of lip motions necessitates a more robust speech encoder for accurate representation. Lastly, humans have diverse speaking styles. A strong style encoder should be designed to extract style representation from an arbitrary style clip.

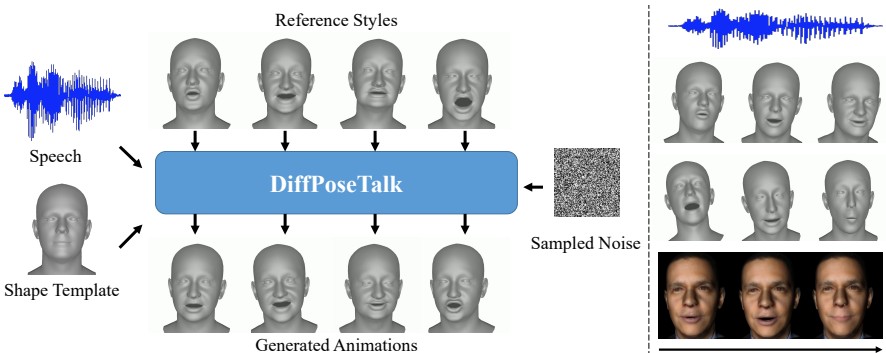

Figure 1: Our proposed DiffPoseTalk system. See text for details.

To address the aforementioned limitations and challenges, we introduce DiffPoseTalk, a novel controllable diffusion-based generative model, to generate high-quality, diverse, speech-matched, and stylistic facial motions for speech-driven 3D facial animation (Figure 1). Our method overcomes the inability of existing diffusion models to be directly transferred to speech-driven expression animation. Compared to existing methods, the main improvement of DiffPoseTalk can be characterized as follows. We use an attention-based architecture to align facial motions with speech, and train a diffusion model to predict the facial expression signal itself (Ramesh et al., 2022b; Tevet et al., 2022) instead of predicting the noise; this architecture allows us to facilitate the subsequent addition of geometric constraints to generate more reasonable results. Along with the expression, we also predict the speaker's head pose and design the corresponding loss function to obtain more natural animations. Furthermore, we exploit Wav2Vec2 (Baevski et al., 2020) to encode the input speech to improve generalization and robustness. Finally, we develop a style encoder to obtain latent style code from a style video clip, and perform classifier-free guidance (Ho & Salimans, 2022b) at inference time to achieve style control. To address the scarcity of co-speech 3D facial animation data by motion capture, we collect and build a speech-driven facial animation dataset with varied speaking styles.

In summary, our contributions are:

- We propose a novel diffusion-based approach to jointly generate diverse and stylistic 3D facial motions with head poses from speech, adequately capturing the many-to-many mapping between speech, style, and motion.

- We develop a style encoder to extract personalized speaking styles from reference videos, which can be used to guide the motion generation in a classifier-free manner.

- We build a high-quality audio-visual dataset that encompasses a diverse range of identities and head poses. This dataset and our code will be made publicly available.

## 2 RELATED WORK

**Speech-Driven 3D Facial Animation.** Existing speech-driven 3D facial animation methods can be roughly divided into procedural and learning-based methods. Procedural approaches generally segment speech into phonemes, which are then mapped to predefined visemes via a set of comprehensive rules. For example, Cohen et al. (2001) uses dominance functions to map phonemes to corresponding facial animation parameters, while Edwards et al. (2016) factors speech-related animation into jaw and lip actions, employing a co-articulation model to animate facial rigs. Although these procedural methods offer explicit control over the resulting animations, they often require intricate parameter tuning and lack the ability to capture the diversity of real-world speaking styles.

Learning-based methods have been growing rapidly over the recent decade. These approaches typically adopt acoustic features like MFCC or pretrained speech model features (Hannun et al., 2014; Baevski et al., 2020) as the speech representation, which is then mapped to 3D morphable model parameters (Zhang et al., 2023a; Peng et al., 2023) or 3D mesh (Cudeiro et al., 2019; Richard et al.,

2021; Fan et al., 2022; Xing et al., 2023) through neural networks. Some methods (Zhang et al., 2023a) also predict head poses in addition to facial animations. However, most existing methods are deterministic and tend to generate over-smoothed lip motions based on the regression models (Cudeiro et al., 2019; Richard et al., 2021; Fan et al., 2022; Xing et al., 2023). CodeTalker (Xing et al., 2023) proposes to use a codebook as a discrete motion prior to achieve accurate lip motions instead of over-smoothed motions, but the approach cannot generate diverse outputs when given the same input.

For those methods that allow stylistic generation (Cudeiro et al., 2019; Fan et al., 2022), they mostly employ one-hot embeddings for training subjects as the style condition, limiting the ability to adapt to new individuals. Imitator (Thambiraja et al., 2023) can adapt to new users but requires subsequent optimization and fine-tuning. In contrast, our method proposed in this paper leverages the strong probabilistic modeling capability of diffusion models, controlled by a style embedding from a novel speaking style encoder in a classifier-free manner to generate diverse and stylistic 3D facial animations.

**Diffusion Probabilistic Models.** Diffusion probabilistic models (Sohl-Dickstein et al., 2015; Ho et al., 2020), which are able to generate high-quality and diverse samples, have achieved remarkable results in various generative tasks (Yang et al., 2022). They leverage a stochastic diffusion process to gradually add noise to data samples, subsequently employing neural architectures to reverse the process and denoise the samples. A key strength of diffusion models lies in their ability to model various forms of distributions and capture complex many-to-many relationships, making them particularly well-suited for our speech-driven facial animation task.

For conditional generation, classifier-guidance (Dhariwal & Nichol, 2021) and classifier-free guidance (Ho & Salimans, 2022a) are widely employed in tasks such as text-to-image synthesis (Rombach et al., 2022), text-to-motion Tevet et al. (2023) synthesis, and audio-driven body motion generation (Zhu et al., 2023; Alexanderson et al., 2023). In particular, diffusion model with classifier-free guidance has achieved impressive results in multi-modal modeling. However, it has gained little attention in speech-driven 3D facial animation tasks.

In this paper, we propose a novel diffusion-based model with a carefully-designed transformer-based denoising network to generate facial motions according to arbitrary speech signals.

## 3 METHOD

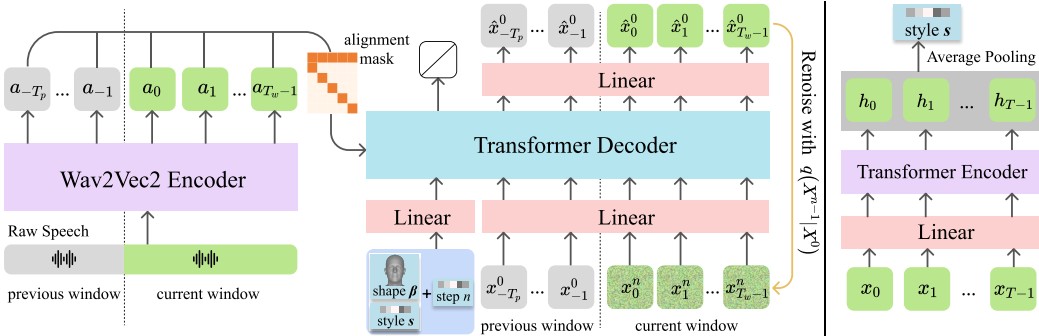

Figure 2: **(Left) Transformer-based denoising network.** We employ a windowing strategy to generate speech-driven 3D facial animations for inputs of arbitrary length. Wav2Vec2-encoded speech features $\boldsymbol{A}_{-T_p:T_w}$, prior clean motion parameters $\boldsymbol{X}^0_{-T_p:0}$, current noisy motion parameters $\boldsymbol{X}^n_{0:T_w}$, shape parameters $\boldsymbol{\beta}$, style feature $\boldsymbol{s}$, and the diffusion timestep $n$ are fed into the transformer decoder. The decoder then predicts clean motion parameters $\hat{\boldsymbol{X}}^0_{-T_p:T_w}$, which are renoised to $\boldsymbol{X}^{n-1}_{0:T_w}$ for the next denoising iteration. **(Right) The speaking style encoder.** The style feature $\boldsymbol{s}$ can be extracted from a sequence of motion parameters $\boldsymbol{X}_{0:T}$ using a transformer encoder.

An overview of our proposed method is illustrated in Figure 2. We adopt a well-established, pre-trained encoder to extract speech features, while using 3DMM as the face representation (Sec-

tion 3.1). A transformer-based denoising network is used for the reverse diffusion process (Section 3.2), where we guide the conditional generation in a classifier-free manner (Section 3.3).

## 3.1 PROBLEM FORMULATION

Our method takes a speech feature $A_{0:T}$[1], a template face shape parameter $\beta$, and a speaking style vector $s$ as input, and generates a 3DMM-based 3D facial animation represented with a sequence of 3DMM expression and pose parameters $X_{0:T}$. The style vector $s$ can be extracted from a short reference video using our speaking style encoder (Section 3.3.1).

**Speech Representation.** In light of recent advancements in 3D facial animation (Fan et al., 2022; Xing et al., 2023; Peng et al., 2023), the state-of-the-art self-supervised pretrained speech model, Wav2Vec2 (Baevski et al., 2020), has proven its capability to yield robust and context-rich audio features that outperforms traditional ones like MFCC. For this reason, we also employ Wav2Vec2 as our speech encoder. Wav2Vec2 is consisted of a temporal convolutional audio feature extractor and a multi-layer transformer encoder. To align the audio features with the facial motions' sampling rate, we introduce a resampling layer after the temporal convolutions. Thus, for a given raw audio clip that matches a facial motion sequence of length $T$, Wav2Vec2 generates a speech representation $A_{0:T}$ that also spans $T$ time steps.

**3D Face Representation.** The FLAME model is a 3D morphable model with $N = 5,023$ vertices and $K = 4$ joints, whose geometry can be represented using parameters $\{\beta, \psi, \theta\}$, where $\beta \in \mathbb{R}^{100}$ is the shape parameter, $\psi \in \mathbb{R}^{50}$ is the expression parameter, and $\theta \in \mathbb{R}^{3K+3}$ is the pose parameter. Given a set of FLAME parameters, the 3D face mesh can be obtained with $M(\beta, \theta, \psi) = W(T_P(\beta, \theta, \psi), \mathbf{J}(\beta), \theta, \mathcal{W})$, where $T_P$ outputs vertices by combining blend shapes, the standard skinning function $W(\mathbf{T}, \mathbf{J}, \theta, \mathcal{W})$ rotates the vertices of $\mathbf{T}$ around joints $\mathbf{J}$, and $\mathcal{W}$ performs linear smoothing. Specifically, we use the shape parameter $\beta$ to serve as the neutral template face for the speaker. For facial animation, we predict the expression parameter $\psi$ as well as the jaw and global rotation components within the pose parameter $\theta$. To simplify notation, we compose $\psi$ and $\theta$ as the "motion parameter" $x$ and rewrite the mesh construction function as $M(\beta, x)$.

## 3.2 FACIAL ANIMATION DIFFUSION MODEL

We propose to use a diffusion model to generate speech-driven facial animation. The diffusion model defines two processes. The forward process is a Markov chain $q(X^n|X^{n-1})$ for $n \in \{1, \ldots, N\}$ that progressively adds Gaussian noise to an initial data sample $X^0$ according to a variance schedule. The original sample is gradually substituted by noises, eventually reaching a standard normal distribution $q(X^N|X^0)$. The reverse process, on the contrary, leverages the distribution $q(X^{n-1}|X^n)$ to recreate the sample from noise. However, this distribution depends on the entire dataset and hence is intractable. Therefore, a denoising network is used to approximate this distribution. In practice, the denoising network is trained to predict the noise (Ho et al., 2020) or the clean sample $X^0$ (Ramesh et al., 2022a). We opt for the latter, as it enables us to incorporate geometric losses that offer more precise constraints on facial motions. The effectiveness of this scheme has been validated by prior works on human body motion generation (Tevet et al., 2023) and our experiments.

### 3.2.1 TRANSFORMER-BASED DENOISING NETWORK

Our transformer-based denoising network, as illustrated in Figure 2, consists of two components: a pretrained Wav2Vec2 encoder for extracting speech features $A$, and a transformer decoder for sampling predicted motions $\hat{X}^0$ from noisy observations $X^n$ ($n = N, N-1, \ldots, 1$) in an iterative manner. A notable design is an alignment mask between the encoder and the decoder, similar to that in Fan et al. (2022), which ensures proper alignment of the speech and motion modalities. Specifically, the motion feature at position $t$ only attends to the speech feature $a_t$. The initial token, which is composed of diffusion timestep $n$ and other conditions, attends to all speech features. We allow the transformer part of the Wav2Vec2 speech encoder to be trainable, which enables Wav2Vec2 to better capture motion information directly from speech. To accommodate sequences of arbitrary lengths, we implement a windowing strategy for the inputs.

---

[1]We use Python style indexing and slicing in this paper, *i. e.*, "$0:T$" includes "$0, 1, \ldots, T-1$".

Formally, the inputs to the denoising network are processed as follows. For a given speech feature sequence of length $T$, we partition it into windows of length $T_w$ (padding is added to the audio if it is not long enough). To ensure seamless transitions between consecutive windows, we include the last $T_p$ frames of speech features $\boldsymbol{A}_{-T_p:0}$ and motion parameters $\boldsymbol{X}^0_{-T_p:0}$ from the preceding window as a conditional input. Note that for the first window, the speech features and motion parameters are replaced with learnable start features $\boldsymbol{A}_{\text{start}}$ and $\boldsymbol{X}_{\text{start}}$. Within each window at diffusion step $n$, the network receives both previous and current speech features $\boldsymbol{A}_{-T_p:T_w}$, the previous motion parameters $\boldsymbol{X}^0_{-T_p:0}$, and the current noisy motion parameters $\boldsymbol{X}^n_{0:T_w}$ sampled from $q\left(\boldsymbol{X}^n_{0:T_w}|\boldsymbol{X}^0_{0:T_w}\right)$. The denoising network then outputs the clean sample as:

$$\hat{\boldsymbol{X}}^0_{-T_p:T_w} = D\left(\boldsymbol{X}^n_{0:T_w}, \boldsymbol{X}^0_{-T_p:0}, \boldsymbol{A}_{-T_p:T_w}, n\right). \tag{1}$$

### 3.2.2 LOSSES

We use the simple loss (Ho et al., 2020) for the predicted sample:

$$\mathcal{L}_{\text{simple}} = \left\|\hat{\boldsymbol{X}}^0_{-T_p:T_w} - \boldsymbol{X}^0_{-T_p:T_w}\right\|^2. \tag{2}$$

To better constrain the generated face motion, we convert the FLAME parameters into zero-head-posed 3D mesh sequences $\boldsymbol{M}_{-T_p:T_w} = M_0\left(\boldsymbol{\beta}, \boldsymbol{X}^0_{-T_p:T_w}\right)$ and $\hat{\boldsymbol{M}}_{-T_p:T_w} = M_0\left(\boldsymbol{\beta}, \hat{\boldsymbol{X}}^0_{-T_p:T_w}\right)$. We then apply the following geometric losses in 3D space: the vertex loss $\mathcal{L}_{\text{vert}}$ (Cudeiro et al., 2019) for the positions of the mesh vertices, the velocity loss $\mathcal{L}_{\text{vel}}$ (Cudeiro et al., 2019) for better temporal consistency, and a smooth loss $\mathcal{L}_{\text{smooth}}$ to penalize large acceleration of the predicted vertices:

$$\mathcal{L}_{\text{vert}} = \left\|\boldsymbol{M}_{-T_p:T_w} - \hat{\boldsymbol{M}}_{-T_p:T_w}\right\|^2, \tag{3}$$

$$\mathcal{L}_{\text{vel}} = \left\|\left(\boldsymbol{M}_{-T_p+1:T_w} - \boldsymbol{M}_{-T_p:T_w-1}\right) - \left(\hat{\boldsymbol{M}}_{-T_p+1:T_w} - \hat{\boldsymbol{M}}_{-T_p:T_w-1}\right)\right\|^2, \tag{4}$$

$$\mathcal{L}_{\text{smooth}} = \left\|\hat{\boldsymbol{M}}_{-T_p+2:T_w} - 2\hat{\boldsymbol{M}}_{-T_p+1:T_w-1} + \hat{\boldsymbol{M}}_{-T_p:T_w-2}\right\|^2. \tag{5}$$

We apply geometric losses $\mathcal{L}_{\text{head}}$ to head motions in a similar way. Refer to the appendix Section A.1 for more details.

In summary, our overall loss is defined as:

$$\mathcal{L} = \mathcal{L}_{\text{simple}} + \lambda_{\text{vert}}\mathcal{L}_{\text{vert}} + \lambda_{\text{vel}}\mathcal{L}_{\text{vel}} + \lambda_{\text{smooth}}\mathcal{L}_{\text{smooth}} + \mathcal{L}_{\text{head}}. \tag{6}$$

### 3.3 STYLE-CONTROLLABLE DIFFUSION MODEL

The facial animation diffusion model is able to generate facial motions conditioned on input speech. We also use speaking style and template face shape as control conditions in addition to the speech. The shape parameter $\boldsymbol{\beta}$ and speaking style feature $\boldsymbol{s}$ are shared across all windows. The denoising network then outputs the clean sample as:

$$\hat{\boldsymbol{X}}^0_{-T_p:T_w} = D\left(\boldsymbol{X}^n_{0:T_w}, \boldsymbol{X}^0_{-T_p:0}, \boldsymbol{A}_{-T_p:T_w}, \boldsymbol{s}, \boldsymbol{\beta}, n\right). \tag{7}$$

### 3.3.1 SPEAKING STYLE ENCODER

We introduce a speaking style encoder designed to capture the unique speaking style of a given speaker from a brief video clip. Speaking style is a multifaceted attribute that manifests in various aspects such as the size of the mouth opening (Cudeiro et al., 2019), facial expression dynamics — especially in the upper face (Xing et al., 2023) — and head movement patterns (Yi et al., 2022; Zhang et al., 2023a). Given the complexity and difficulty in quantitatively describing speaking styles, we opt for an implicit learning approach through contrastive learning. We operate under the assumption that the short-term speaking styles of the same person at two proximate times should be similar.

**Architecture.** The speaking style encoder (see Figure 2) utilizes a transformer encoder to extract style features from a sequence of motion parameters $\boldsymbol{X}_{0:T}$. The encoder features $\{\boldsymbol{h}_i\}$ are aggregated by average pooling into the style embedding $\boldsymbol{s}$. Formally, this is described as:

$$\boldsymbol{s} = SE\left(\boldsymbol{X}_{0:T}\right). \tag{8}$$

**Contrastive Training.** We use the NT-Xent loss (Chen et al., 2020) for contrastive learning. Each training minibatch consists of $N_s$ samples of speech features and motion parameters of length $2T$. We split the sample length in half to get $N_s$ pairs of positive examples. Given a positive pair, the other $2(N_s - 1)$ examples are treated as negative examples. We use cosine similarity as the similarity function. The loss function for a positive pair examples $(i, j)$ is defined as:

$$\mathcal{L}_{i,j} = -\log \frac{\exp\left(\text{cos\_sim}(\boldsymbol{s}_i, \boldsymbol{s}_j)/\tau\right)}{\sum_{k=1}^{2N_s} \mathbf{1}_{k \neq i} \exp\left(\text{cos\_sim}(\boldsymbol{s}_i, \boldsymbol{s}_k)/\tau\right)}, \tag{9}$$

where $\mathbf{1}_{k \neq i}$ is an indicator function and $\tau$ represents a temperature parameter. The overall loss is computed across all positive pairs for both $(i, j)$ and $(j, i)$.

### 3.3.2 Training Strategy

In the framework of our window-based generation approach, our network faces two different scenarios: (a) generating the initial window, where the previous window conditions are learnable start features, and (b) generating subsequent windows, where the conditions are speech features and motions parameters from the preceding window. The network also requires a shape parameter $\boldsymbol{\beta}$ and a speaking style feature $\boldsymbol{s}$ in both scenarios. Therefore, we propose a novel training strategy to meet this demand. Specifically, each training sample comprises a frame of shape parameter $\boldsymbol{\beta}$, a speech clip (which will be encoded into speech features $\boldsymbol{A}_{0:2T_w}$ by the Wav2Vec2 encoder), and a corresponding motion parameter sequence $\boldsymbol{X}_{0:2T_w}^0$. We partition the sample into two windows and employ the speaking style encoder to derive style features for each, resulting in $(\boldsymbol{s}_a, \boldsymbol{s}_b)$. The tuple $(\boldsymbol{A}_{0:T_w}, \boldsymbol{X}_{0:T_w}^0, \boldsymbol{s}_b)$ is used to train the first window, while $(\boldsymbol{A}_{T_w:2T_w}, \boldsymbol{X}_{T_w:2T_w}^0, \boldsymbol{s}_a)$ with the previous window conditions $(\boldsymbol{A}_{T_w-T_p:T_w}, \boldsymbol{X}_{T_w-T_p:T_w}^0)$ is used to train the second window. Taking into account that the actual length of speech during generation may not fully occupy the window, we introduce a random truncation of samples during training. This approach ensures that the model is robust to variations in speech length.

### 3.3.3 Sampling with Incremental Classifier-Free Guidance

During generation, we sample the result $\boldsymbol{X}^0$ conditioned on $(\boldsymbol{A}, \boldsymbol{s}, \boldsymbol{\beta})$ in an iterative manner. Specifically, we estimate the clean sample as $\hat{\boldsymbol{X}}^0 = D\left(\boldsymbol{X}^n, \boldsymbol{A}, \boldsymbol{s}, \boldsymbol{\beta}, n\right)$ and subsequently reintroduce noise to obtain $\boldsymbol{X}^{n-1}$. This process is repeated for $n = N, N-1, \ldots, 1$, in accordance with the process in Tevet et al. (2023).

Furthermore, we find it beneficial to apply classifier-free guidance (Ho & Salimans, 2022a) with respect to both the speech and style conditions. We experimented with the incremental scheme which has been successfully applied to image generation from both input images and text instructions (Brooks et al., 2023), where

$$\hat{\boldsymbol{X}}^0 = D\left(\boldsymbol{X}^n, \emptyset, \emptyset, \boldsymbol{\beta}, n\right) + w_a \left[D\left(\boldsymbol{X}^n, \boldsymbol{A}, \emptyset, \boldsymbol{\beta}, n\right) - D\left(\boldsymbol{X}^n, \emptyset, \emptyset, \boldsymbol{\beta}, n\right)\right] \\ + w_s \left[D\left(\boldsymbol{X}^n, \boldsymbol{A}, \boldsymbol{s}, \boldsymbol{\beta}, n\right) - D\left(\boldsymbol{X}^n, \boldsymbol{A}, \emptyset, \boldsymbol{\beta}, n\right)\right]. \tag{10}$$

The $w_a$ and $w_s$ are the guidance scales for audio and style, respectively. During training, we randomly set the style condition to $\emptyset$ with 0.45 probability, and set both the audio and style conditions to $\emptyset$ with 0.1 probability.

## 4 Experiments

### 4.1 Datasets

Different from closely related works (Cudeiro et al., 2019; Richard et al., 2021; Fan et al., 2022; Xing et al., 2023) that use 3D mesh face data for training, our approach leverages the widely used 3DMM — specifically FLAME (Li et al., 2017) — for face representation. Our rationale for this choice is twofold. First, given the computational intensity of diffusion models, the lower-dimensional 3DMM parameters offer a substantial advantage in terms of computational speed when compared to predicting mesh vertices. Second, capturing real-world 3D mesh data requires professional motion capture systems and considerable investments of time and effort, thereby constraining the scale and diversity

of data that can be collected. For example, VOCASET (Cudeiro et al., 2019) provides less than 30 minutes of data from just 12 subjects. Conversely, 2D audio-visual datasets (Afouras et al., 2018; Zhang et al., 2021) are more accessible and afford far greater coverage, which can include thousands of speakers and tens of thousands of sentences across diverse speaking styles. Utilizing a 3D face reconstruction method (Filntisis et al., 2022) featuring accurate reconstruction of lip movements, we can compile a rich set of 3DMM parameters with precise lips based on these 2D datasets.

We use a combination of two datasets in this work. First, we incorporate the High-Definition Talking Face (HDTF) dataset (Zhang et al., 2021), a high-quality audio-visual collection featuring videos in 720P-1080P resolution sourced from YouTube. Using the downloading script provided by the authors, we successfully acquired 348 videos, covering 246 subjects and yielding approximately 16 hours of footage. However, these videos are mainly collected from three political weekly address programs, resulting in a dataset biased towards formal speaking styles with limited facial expressions and head movements. To address this limitation, we introduce an additional dataset — Talking Face with Head Poses (TFHP) — which comprises 704 videos of 342 subjects, totaling 10.5 hours. Our TFHP dataset is more diversified in content, featuring video clips from lectures, online courses, interviews, and news programs, thereby capturing a wider array of speaking styles and head movements.

We split the combined dataset by speakers, resulting in 460 for training, 64 for validation, and 64 for testing. All videos are converted to 25 fps. A state-of-the-art 3D face reconstruction method (Filntisis et al., 2022) featuring the accurate reconstruction of lip movements is adopted to convert the 2D video dataset into 3DMM parameters.

## 4.2 Implementation Details

We use a four-layer transformer encoder with four attention heads for the speaking style encoder, with feature dimension $d_s = 128$, sequence length $T = 100$ (4 seconds), and temperature $\tau = 0.1$. We train the encoder with the Adam optimizer (Kingma & Ba, 2015) for 26k iterations, with a batch size of 32 and a learning rate of $1\mathrm{e}{-4}$.

We use an eight-layer transformer decoder with eight attention heads for the denoising network, with the feature dimension $d = 512$, the window length $T_w = 100$, and $T_p = 10$. We adopt a cosine noise schedule with diffusion $N = 500$ steps. We train the denoising network with the Adam optimizer for 90k iterations, using 5k warmup steps, batch size 16, and learning rate $1\mathrm{e}{-4}$. We set $\lambda_{\mathrm{vert}} = 2\mathrm{e}6$, $\lambda_{\mathrm{vel}} = 1\mathrm{e}7$, and $\lambda_{\mathrm{smooth}} = 1\mathrm{e}5$ to balance the magnitude of the losses. The overall training on an Nvidia 3090 GPU takes about 12 hours.

## 4.3 Comparison with States of the Arts

We compare our approach with two state-of-the-art 3D facial animation methods: FaceFormer (Fan et al., 2022) and CodeTalker (Xing et al., 2023). Both methods employ 3D mesh as their face representation. Thus, we train them with 3D mesh data generated from the reconstructed 3DMM parameters. Recognizing the scarcity of speech-driven 3D animation methods that account for head poses, we also compare with two 2D talking face generation methods, Yi et al. (2022) and SadTalker, which incorporate head movements and utilize a 3DMM as an intermediate facial representation.

**Quantitative Evaluation.** Following previous studies, we employ two established metrics — lip vertex error (LVE) (Richard et al., 2021) and upper face dynamics deviation (FDD) (Xing et al., 2023) — for the quantitative evaluation of generated facial expressions. LVE measures lip synchronization by calculating the maximum L2 error across all lip vertices for each frame. FDD evaluates the upper face motions, which are closely related to speaking styles, by comparing the standard deviation of each upper face vertex's motion over time between the prediction and ground truth. To assess head motion, we use beat alignment (BA) (Li et al., 2022; Zhang et al., 2023b), albeit with a minor modification: we compute the synchronization of detected head movement beats between the predicted and actual outcomes. For examining the diversity of expressions and poses generated from *identical* input, we follow Ren et al. (2022) to compute a diversity score. Since the size of the mouth opening can also indicate speaking style (Cudeiro et al., 2019), we introduce a new metric called mouth opening difference (MOD), which measures the average difference in the size of the mouth opening between the prediction and ground truth.

Table 1: Quantitative evaluation of the comparative methods, our proposed method, and ablation study variants. We run the evaluation 10 times and report the average score with a $95\%$ confidence interval. The BA metric does not apply to FaceFormer and CodeTalker, as both of the methods do not generate head poses. The vertex-related metrics are not comparable with SadTalker due to its different face topology.

| Methods | LVE (mm) ↓ | FDD ($\times 10^{-5}$m) ↓ | MOD (mm) ↓ | BA ↑ |
|---|---|---|---|---|
| FaceFormer | $9.85^{\pm 0.019}$ | $18.07^{\pm 0.013}$ | $2.30^{\pm 0.013}$ | N/A |
| CodeTalker | $11.78^{\pm 0.043}$ | $15.19^{\pm 0.050}$ | $3.00^{\pm 0.031}$ | N/A |
| Yi et al. | $10.35$ | $21.85$ | $2.52$ | $0.237$ |
| SadTalker | — | — | — | $0.236^{\pm 0.001}$ |
| Ours w/o $\mathcal{L}_{geo}$ | $11.60^{\pm 0.012}$ | $15.53^{\pm 0.078}$ | $2.45^{\pm 0.006}$ | $0.302^{\pm 0.002}$ |
| Ours w/o AM | $16.06^{\pm 0.017}$ | $13.73^{\pm 0.057}$ | $\mathbf{1.99^{\pm 0.011}}$ | $0.240^{\pm 0.001}$ |
| Ours w/o CFG | $9.75^{\pm 0.013}$ | $10.61^{\pm 0.029}$ | $2.37^{\pm 0.014}$ | $0.298^{\pm 0.002}$ |
| Ours w/o SSE | $10.83^{\pm 0.011}$ | $11.79^{\pm 0.053}$ | $2.12^{\pm 0.016}$ | $0.298^{\pm 0.001}$ |
| Ours | $\mathbf{9.44^{\pm 0.008}}$ | $\mathbf{10.56^{\pm 0.026}}$ | $2.07^{\pm 0.011}$ | $\mathbf{0.305^{\pm 0.002}}$ |

Table 2: The diversity score.

| Method | Diversity ↑ | |
|---|---|---|
| | Exp | Head Pose |
| FaceFormer | 0 | N/A |
| CodeTalker | 0 | N/A |
| Yi et al. | 0 | 0 |
| SadTalker | 0 | 0.788 |
| Ours | **1.016** | **1.069** |

Table 3: User study results.

| Method | Lip Sync ↑ | Style Sim ↑ | Natural ↑ |
|---|---|---|---|
| FaceFormer | 1.54 | 1.59 | 1.55 |
| CodeTalker | 1.92 | 1.74 | 1.92 |
| Ours (no HP) | **2.61** | **2.54** | **2.54** |
| Yi et al. | 1.30 | 1.37 | 1.35 |
| SadTalker | 1.85 | 1.70 | 1.79 |
| Ours | **2.74** | **2.72** | **2.73** |

We present the quantitative results in Table 1 and the diversity scores in Table 2. Our method outperforms all others across all metrics, achieving the best lip synchronization and head pose beat alignment. Additionally, the FDD, MOD, and BA metrics suggest that our method most effectively captures speaking styles. As for diversity, the other methods employ deterministic approaches for motion generation, with the exception of SadTalker, which samples head poses from a VAE. Consequently, these methods are unable to produce varied expressions and head poses from identical inputs, falling short in capturing this many-to-many mapping.

**Qualitative Evaluation.** We show the comparison of our method with other comparative methods in Figure 3. Our method excels in handling challenging cases, such as articulating bilabial consonants and rounded vowels. Moreover, the generated results have the closest speaking style to the ground truth in aspects like upper facial expressions and mouth opening size. Notably, our approach can also spontaneously produce random movements like eye blinks, which are implicitly learned from the data distribution.

**User Study.** To conduct a more comprehensive assessment of our approach, we designed a user study with the following experiment setup. Several methods are categorized into two groups based on whether head motion are generated. The group without head motion includes FaceFormer, CodeTalker, and our method (with head pose set to zero) and consists of 15 sets of questions. The group with head motion involves Yi et al. (2022), SadTalker, and our approach, comprising 10 sets of questions. In each set, participants are shown the ground truth animation as well as animations generated by each method. Participants are then asked to rate the lip synchronization, similarity to the speaking style of the ground truth, and the naturalness of facial expressions and head movements, on a scale of 1-3. Twenty-five participants took part in the study, and the results are presented in Table 3. The findings demonstrate that our method significantly outperforms existing works in

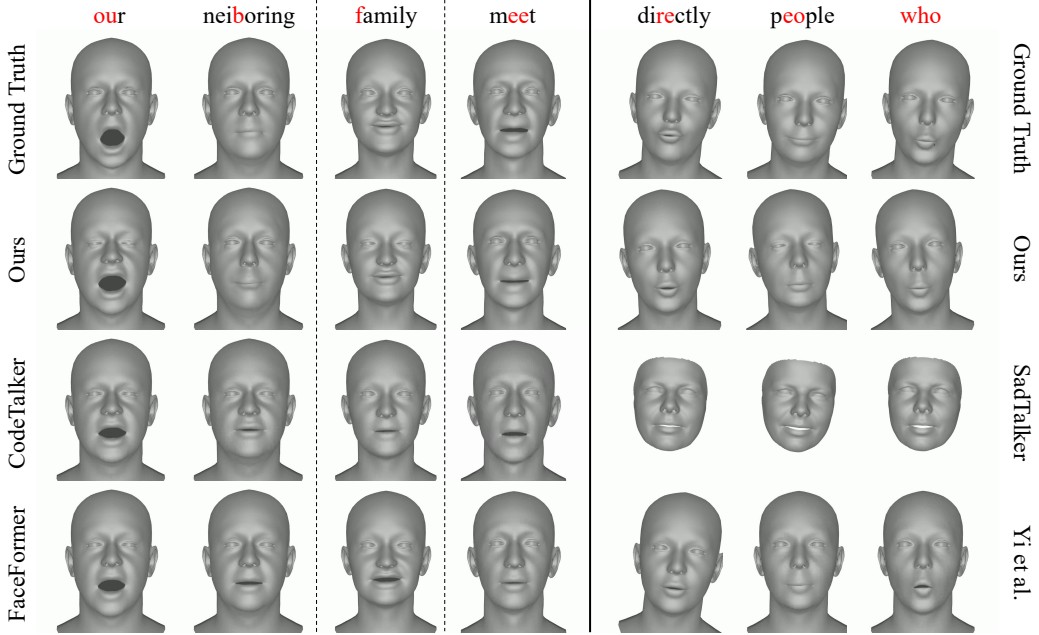

Figure 3: Qualitative comparison of our method with state of the arts. See supplementary demo video for more details.

terms of lip synchronization, similarity to the ground truth speaking style, and the naturalness of facial expressions and head movements.

**Ablation Study.** We conduct ablation experiments to evaluate the impact of key components and techniques in our proposed method: 1) Ours w/o SSE: We remove the speaking style encoder. 2) Ours w/o $L_{geo}$: All geometric losses are removed, including those related to facial vertices and head motions. 3) Ours w/o AM: The alignment mask between the encoder and decoder is removed. 4) Ours w/o CFG: We do not employ classifier-free guidance. The results are displayed in Table 1. As can be observed, the removal of the speaking style encoder leads to a decline in performance across all metrics, as our method is no longer capable of generating animations tailored to the speaking style of individuals. Removing all geometric losses results in our method being unable to produce precise facial animations. Removing the alignment mask causes a serious out-of-sync problem. The exclusion of classifier-free guidance also affects the quality of the results.

## 5 CONCLUSION AND FUTURE WORK

Speech-driven expression animation has a wide range of applications in daily life and has received extensive attention from researchers. It involves a challenging many-to-many mapping across modalities between speech and expression animation. In this paper, we present DiffPoseTalk, a novel diffusion-based method for generating diverse and stylistic 3D facial animations and head poses from speech. We leverage the capability of the diffusion model to effectively replicate the distribution of diverse forms, thereby addressing the many-to-many mapping challenge. Additionally, we resolve the limitations associated with current diffusion models that hinder its direct application to speech-driven expression animation. Leveraging the power of the diffusion model and the speaking style encoder, our approach excels in capturing the many-to-many relationships between speech, style, and motion. We also present a high-quality audio-visual dataset containing 704 videos covering different speaking styles with rich head movements and trained our model based on it. Experiment results demonstrate that our method is effective in capturing speaking styles and outperforms prior state-of-the-art methods in terms of lip synchronization, head pose beat alignment, and naturalness. In future work, we will consider accelerating the generation process.

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

# A    APPENDIX

## A.1    GEOMETRY LOSSES FOR HEAD MOTIONS

Similarly, we apply losses $\mathcal{L}_{\text{head\_ang}}$, $\mathcal{L}_{\text{head\_vel}}$, and $\mathcal{L}_{\text{head\_smooth}}$ to the head pose component $p$ of the pose parameter $\theta$ to constrain head movements:

$$\mathcal{L}_{\text{head\_ang}} = \left\| \boldsymbol{P}_{-T_p:T_w} - \hat{\boldsymbol{P}}_{-T_p:T_w} \right\|^2, \tag{11}$$

$$\mathcal{L}_{\text{head\_vel}} = \left\| \left( \boldsymbol{P}_{-T_p+1:T_w} - \boldsymbol{P}_{-T_p:T_w-1} \right) - \left( \hat{\boldsymbol{P}}_{-T_p+1:T_w} - \hat{\boldsymbol{P}}_{-T_p:T_w-1} \right) \right\|^2, \tag{12}$$

$$\mathcal{L}_{\text{head\_smooth}} = \left\| \hat{\boldsymbol{P}}_{-T_p+2:T_w} - 2\hat{\boldsymbol{P}}_{-T_p+1:T_w-1} + \hat{\boldsymbol{P}}_{-T_p:T_w-2} \right\|^2. \tag{13}$$

Furthermore, we discover that constraining the velocity and acceleration of head movement at the start of the current window to match those at the end of the previous window helps with generating smooth transition and prevents abrupt changes in head posture. Thus, we define the pose sequence during the transition as $\bar{\boldsymbol{P}}_{-3:3} = \{\boldsymbol{P}_{-3:0}, \hat{\boldsymbol{P}}_{0:3}\}$ and the transition loss $\mathcal{L}_{\text{trans}}$ as:

$$\begin{aligned}
\mathcal{L}_{\text{head\_trans}} = &\left\| \left( \bar{\boldsymbol{P}}_{0:2} - \bar{\boldsymbol{P}}_{-1:1} \right) - \left( \bar{\boldsymbol{P}}_{-1:1} - \bar{\boldsymbol{P}}_{-2:0} \right) \right\|^2 + \\
&\left\| \left( \bar{\boldsymbol{P}}_{0:3} - 2\bar{\boldsymbol{P}}_{-1:2} + \bar{\boldsymbol{P}}_{-2:1} \right) - \left( \bar{\boldsymbol{P}}_{-1:2} - 2\bar{\boldsymbol{P}}_{-2:1} + \bar{\boldsymbol{P}}_{-3:0} \right) \right\|^2.
\end{aligned} \tag{14}$$

The overall head loss is expressed as:

$$\mathcal{L}_{\text{head}} = \lambda_{\text{head\_ang}}\mathcal{L}_{\text{head\_ang}} + \lambda_{\text{head\_vel}}\mathcal{L}_{\text{head\_vel}} + \lambda_{\text{head\_smooth}}\mathcal{L}_{\text{head\_smooth}} + \lambda_{\text{head\_trans}}\mathcal{L}_{\text{head\_trans}}. \tag{15}$$

During training, the weights are set as: $\lambda_{\text{head\_ang}} = 0.05$, $\lambda_{\text{head\_vel}} = 5$, $\lambda_{\text{head\_smooth}} = 0.5$, and $\lambda_{\text{head\_trans}} = 0.5$.

## A.2    EVALUATION DETAILS

We segment the test videos into clips ranging from 5 to 16 seconds for evaluation. Among the compared methods, FaceFormer and CodeTalker require a one-hot training speaker label as a style condition. While prior studies evaluated these methods using all available speaker labels, such an approach is infeasible in our case due to the fact that our dataset is comprised of 460 training identities (in contrast, the VOCASET has only 8 training identities). Consequently, we randomly select labels for each clip. Similarly, SadTalker requires a head pose label chosen from 46 styles, which is also set randomly. Note that we use the pretrained model provided by the authors, as the training code is not publicly available. For both Yi et al. (2022) and our approach, we utilize the style features extracted from adjacent clips of the same video for each evaluated clip. The classifier free guidance scales are set to $w_a = 1.15$ and $w_s = 2.5$.

## A.3    INFERENCE SPEED

When conducting inference with an Intel Xeon Gold 5218R CPU and a Nvidia 3090 GPU, Diff-PoseTalk can generate the motion parameters at 30 FPS. Using the style encoder to extract a style feature from a reference segment takes only 2ms.

For live streaming scenarios, our system introduces a 7.33-second delay. This delay is primarily due to our windowing strategy, which involves an initial 4 seconds wait to fill the first window and an additional 3.33 seconds for processing. However, our inference does not introduce any further latency once this initial processing is complete.

## A.4    VISUALIZATION OF THE STYLE AND CONTENT DISENTANGLEMENT

To examine the disentanglement of our generation's style and content, we randomly selected 10 speaking styles and 20 audio clips to generate $10 \times 20$ animations. The speaking styles from these animations are then extracted using our style encoder. We employ t-SNE (Van der Maaten & Hinton, 2008) for visualizing these extracted speaking styles. The result in Figure 4 demonstrates that animations with identical reference styles are clustered together, regardless of the content differences.

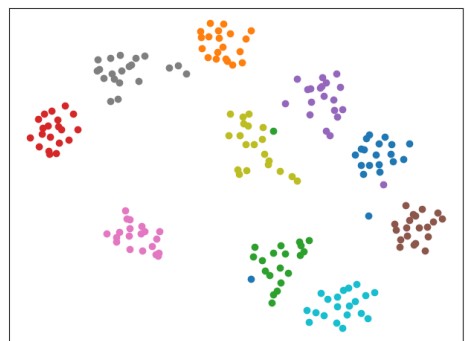

Figure 4: Visualization of the disentanglement of style and content. Colors indicate different styles. Animations with identical reference styles are clustered together, regardless of the content differences.

| Method | FaceFormer | CodeTalker | Ours* |
|---|---|---|---|
| LVE (mm) ↓ | 4.43 | 4.55 | **4.35** |
| FDD ($\times 10^{-5}$m) ↓ | 13.61 | 10.34 | **8.85** |

Table 4: Quantitative evaluation of the comparative methods and our proposed method on VOCASET. Ours*: Head movement prediction is removed and the style encoder is replaced with one-hot encoding.

### A.5    USER STUDY

We provide screenshots of the user study system and samples in Figure 5.

### A.6    EXPERIMENTS ON VOCASET

Our method is designed for generating style-referenced 3D face animations with head movements. However, the cleaned VOCASET lacks head poses and has very limited coverage of identities and speaking styles. Additionally, it does not provide corresponding 3DMM parameters. To train and evaluate our method on VOCASET, we take the following measures: First, we use an optimization-based method[2] to convert 3D meshes into 3DMM parameters. Second, because our style encoder requires a relatively large number of distinct styles for contrastive training—a demand VOCASET cannot meet—we resort to traditional one-hot style encoding. Third, we remove the head movements prediction. Finally, we set $T_w = 50$ and $T_p = 10$ to accommodate the relatively short clip duration in VOCASET. We follow FaceFormer (Fan et al., 2022) and CodeTalker (Xing et al., 2023) to train our method on resampled data at 30 fps.

For fair comparison, we train FaceFormer and CodeTalker on 3D meshes converted from the 3DMM parameters, while keeping other settings consistent with the original literature. The LVE and FDD metrics used in CodeTalker are adopted for quantitative evaluation. Our method outperforms other baselines as presented in Table 4. Please refer to the demo video for qualitative evaluations.

### A.7    ETHICAL DISCUSSIONS

Since our approach is able to generate realistic talking head sequences, there are risks of misusing, such as deepfake generation and deliberate manipulation. Therefore, we firmly require that all talking face sequences generated by our method be marked or noted as synthetic data. Moreover, we will make our code publicly available to the deepfake detection community to further ensure that our proposed method can be applied positively. We also hope to raise awareness of the risks and support regulations preventing technology abuse involving synthetic videos.

---

[2] https://github.com/TimoBolkart/voca/blob/master/compute_FLAME_params.py

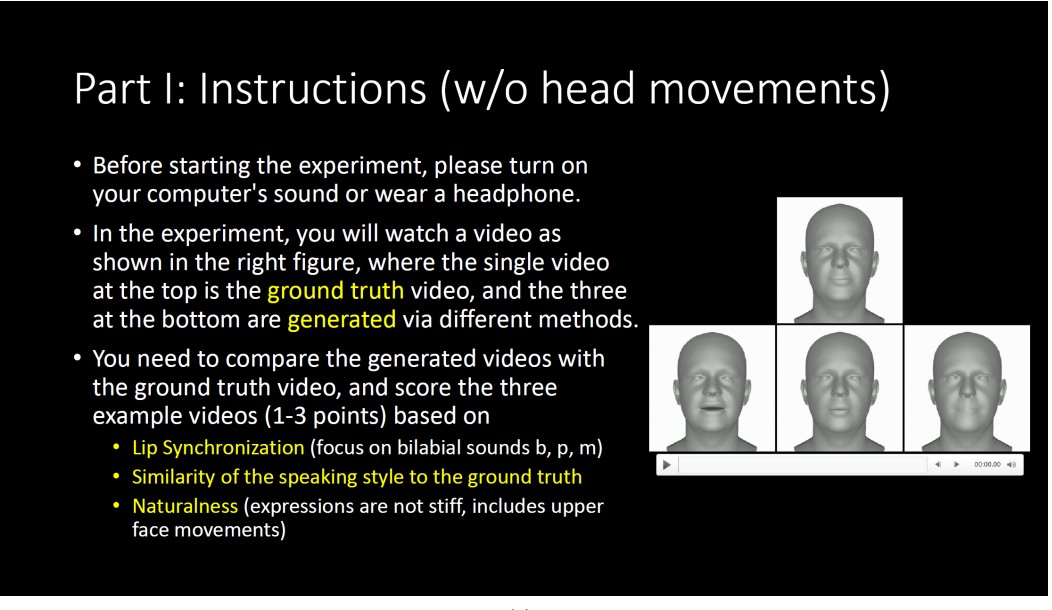

(a)

(b)

Figure 5: Screenshots of the user study system and samples.

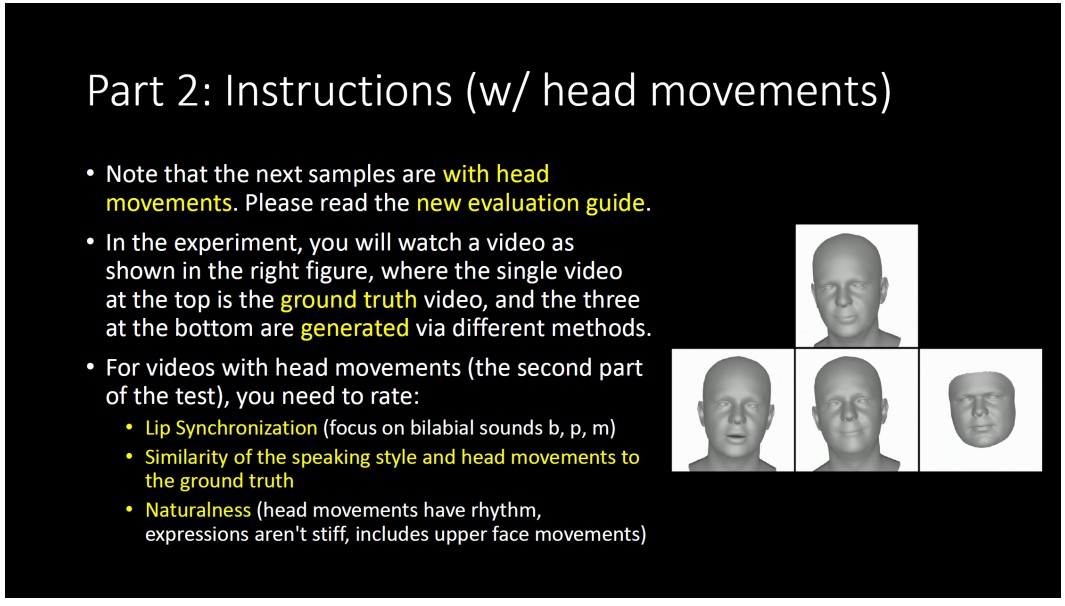

(c)

(d)

Figure 5: Screenshots of the user study system and samples.

## A.8 LIMITATIONS

Although our method is able to generate high-quality stylistic 3D talking face animation with vivid head poses, there are still some limitations within our framework that could be addressed in follow-up works. Firstly, the computational cost of inference is relatively high due to the sequential nature of the denoising process. To mitigate this, future research can explore more advanced denoisers such as DPM-solver++ (Lu et al., 2022). Secondly, like existing SOTAs, our approach focuses on animating the face shape while ignoring the inner mouth (including teeth and tongue). Exploring representation and animation of the inner mouth can lead to more realistic results. Lastly, a promising direction for future research would be to collect real-world 3D talking data that encompasses a broader range of identities and styles, which would further enhance the effectiveness of our approach and contributes to the research community.

