# OpenReview forum: "DiffPoseTalk: Speech-Driven Stylistic 3D Facial Animation and Head Pose Generation via Diffusion Models"
_ICLR.cc/2024/Conference — Submitted to ICLR 2024_

### Official Review · Reviewer_rH7c · 2023-10-31

**Soundness:** 4 excellent
**Presentation:** 3 good
**Contribution:** 2 fair
**Rating:** 6
**Confidence:** 4

**Summary:**

The paper presents a diffusion-based method to generate co-speech 3D facial motions, particularly motions conditioned on various speaking styles. To this end, the proposed learning model consists of multiple components: the Wav2Vec2 Encoder to encode the speech, a transformer-based denoising decoder to generate current face motion parameters from the speech embeddings and noisy, past face motion parameters, and a transformer-based style encoder trained on a set of reference videos with a contrastive loss to incorporate the desired speaking styles into the face motions. The authors show the benefits of their proposed method through quantitative evaluations, ablation studies, and user studies.

**Strengths:**

1. The proposed method of generating facial motion parameters from speech through a denoising diffusion process is technically sound.

2. The style encoder offers customizability to the generated face motions and improves the utility of the proposed method.

**Weaknesses:**

1. In Sec. 1, para 2, the authors note that facial motions require more "precise" alignment with speech than other human body motions, such as gestures and dance. However, it is not clear what this "precision" entails and how, or if, it can be measured. What are the specific design choices in the denoising diffusion architecture, or the training loss functions and hyperparameters, or some other aspects, that are necessary for the successful learning of facial motions from speech? In other words, why would an existing diffusion architecture for body motion generation (such as [A] or [B], albeit with different training features) not work for this problem?

[A] Ao, Tenglong, Zeyi Zhang, and Libin Liu. "GestureDiffuCLIP: Gesture diffusion model with CLIP latents." ACM Transactions on Graphics, August 2023.
[B] Dabral, Rishabh, Muhammad Hamza Mughal, Vladislav Golyanik, and Christian Theobalt. "Mofusion: A framework for denoising-diffusion-based motion synthesis." In Proceedings of the IEEE/CVF Conference on Computer Vision and Pattern Recognition, pp. 9760-9770. 2023.

2. While the style encoder offers more customizability to the generated face motions, it seems to be limited to the reference videos received during training. Is this understanding correct, or can the style encoder generalize to novel styles during inference?

**Questions:**

1. What is the latency of the end-to-end generation pipeline during inference? What is the latency of the stylization component?

2. For Eqn. 2, is there any specific reason to operate on the vertex space and on the parameter space? The vertex space is much higher dimensional and less constrained, which could make the training more unstable.

---

> ### Author Response · Authors · 2023-11-20
> **Reply to Reviewer rH7c**
>
> Thank you for the valuable comments.
>
> Q1: In Sec. 1, para 2, the authors note that facial motions require more "precise" alignment with speech than other human body motions, such as gestures and dance. However, it is not clear what this "precision" entails and how, or if, it can be measured. What are the specific design choices in the denoising diffusion architecture, or the training loss functions and hyperparameters, or some other aspects, that are necessary for the successful learning of facial motions from speech? In other words, why would an existing diffusion architecture for body motion generation (such as [A] or [B], albeit with different training features) not work for this problem?
>
> A1: This precision refers to the need for accurate synchronization between the articulation of sounds and the corresponding lip movements. In contrast, gestures can have more flexibility in their timing, allowing them to occur slightly before or after the associated speech. Also, gestures can be less specific in their correlation with speech. As shown in [1], the correlation coefficient between voice pitch and jaw is 0.69, whereas with other body motions it is below 0.42.
>
> As stated in Sec. 1, Para. 2, lip motions are semantically rich (can indicate the speech articulation compared to gestures and dancing). Thus, speech-driven facial animations require stronger speech encoder to extract phoneme-level features. Furthermore, it requires specific designs to align generated motions precisely to speech articulations. However, existing diffusion architectures for body motion generation typically lack these designs. For example, neither [A] nor [B] employ encoders for audio processing (they use traditional acoustic features), and neither have a design that aligns local motion features with local audio feature. To this end, we adapted the well-established transformer encoder-decoder structure, which was widely used in previous non-diffusion-based speech-driven facial animation works, to our diffusion model. The encoder, which was a pretrained Wav2Vec2 encoder, provided robust and contextualized speech features. The alignment mask serves as a crucial component to align the local speech features with the local motion features from the decoder. As the ablation study indicated, removing the alignment mask significantly degraded lip synchronization (from 9.44 to 16.06).
>
> - [1] M. Berry, S. Lewin, and S. Brown, “Correlated expression of the body, face, and voice during character portrayal in actors”, Scientific Reports, vol. 12, no. 1, p. 8253, 2022.
>
> Q2: While the style encoder offers more customizability to the generated face motions, it seems to be limited to the reference videos received during training. Is this understanding correct, or can the style encoder generalize to novel styles during inference?
>
> A2: The style encoder can generalize to novel styles during inference. In fact, all the experiments and examples in this paper used novel videos as the reference videos.
>
> Q3: What is the latency of the end-to-end generation pipeline during inference? What is the latency of the stylization component?
>
> A3: Although our method can inference at 30 fps, for live streaming scenarios, it introduces a 7.33-second delay. This delay is primarily due to our windowing strategy, which involves an initial 4 seconds wait to fill the first window and an additional 3.33 seconds for processing. However, once this initial processing is complete, our inference does not introduce any further latency. Regarding the style encoder, inference on a reference segment takes just 2ms, and the style feature can actually be extracted beforehand. We have added these discussions to Sec. A.3 of the revised appendix.
>
> Q4: For Eqn. 2, is there any specific reason to operate on the vertex space and on the parameter space? The vertex space is much higher dimensional and less constrained, which could make the training more unstable.
>
> A4: Actually, Eqn. 2 (the simple loss), calculates the MSE of the predicted results and the ground truth in the parameter space instead of in the vertex space. We only calculate the geometric losses in the vertex space because the vertex space can yield better loss guidance for the position, velocity, and acceleration of face vertices than the parameter space.

---

> > ### Comment · Reviewer_rH7c · 2023-11-22
> > **Response to authors**
> >
> > I thank the authors for their responses, which address my concerns and questions. I maintain my original score.

---

### Official Review · Reviewer_3BSS · 2023-10-31

**Soundness:** 3 good
**Presentation:** 4 excellent
**Contribution:** 3 good
**Rating:** 8
**Confidence:** 4

**Summary:**

The paper proposes a framework DiffPoseTalk that utilizes a diffusion model with a style encoder. DiffPoseTalk generates 3D speech-driven talking face videos and it is capable of changing the style of the generated videos and can generate video depicting the style of the given short reference video. The model uses a pre-trained Wav2Vec2 encoder as an audio feature extractor and 3DMM as face representation. Also, it uses a transformer-based denoising network for 3D talking face generation. Moreover, they built a talking face dataset TFHP with 704 videos of 302 subjects.

**Strengths:**

A talking face dataset is proposed.

The manuscript is clear and easy to follow.

The manuscript has equations and figures that support the writing.

The model is evaluated well and it is compared to state-of-the-art (see question 1)

**Weaknesses:**

There is no discussion of ethical considerations.

Limitations are not elaborated.

**Questions:**

The ablation study model without CFG shows that it does not have much effect on the results (especially LVE and FDD). Can you elaborate this?

---

> ### Author Response · Authors · 2023-11-20
> **Reply to Reviewer 3BSS**
>
> Thank you for the valuable comments.
>
> Q1: Discussion of ethical considerations.
>
> A1: We have added the following discussion to Sec. A7 of the revised appendix.
> “Since our approach is able to generate realistic talking head sequences, there are risks of misusing, such as deepfake generation and deliberate manipulation. Therefore, we firmly require that all talking face sequences generated by our method be marked or noted as synthetic data. Moreover, we will make our code publicly available to the deepfake detection community to further ensure that our proposed method can be applied positively. We also hope to raise awareness of the risks and support regulations preventing technology abuse involving synthetic videos.”
>
> Q2: Description of limitations.
>
> A2: The following discussions on limitations have been added to the revised appendix.
> “Although our method is able to generate high-quality stylistic 3D talking face animation with vivid head poses, there are still some limitations within our framework that could be addressed in follow-up works. Firstly, the computational cost of inference is relatively high due to the sequential nature of the denoising process. To mitigate this, future research can explore more advanced denoisers such as DPM-solver++. Secondly, like existing SOTAs, our approach focuses on animating the face shape while ignoring the inner mouth (including teeth and tongue). Exploring representation and animation of the inner mouth can lead to more realistic results.  Lastly, a promising direction for future research would be to collect real-world 3D talking data that encompasses a broader range of identities and styles, which would further enhance the effectiveness of our approach and contributes to the research community.”
>
> Q3: Discussion on the ablation study model without CFG.
>
> A3: We hypothesize that the relatively small improvement may arise from two reasons: (1) Classifier-free guidance is a technique in diffusion models that aims to reduce the diversity of generated samples while enhancing the quality of each individual sample. Unlike text-to-image problem, our task exhibits relatively less variation with respect to speech, style, and motion. Consequently, the trade-off between diversity and the quality of each individual sample becomes less apparent. Therefore, the effectiveness of classifier-free guidance in this context might be somewhat limited. (2) There might be a mild train-test mismatch issue during the CFG process, where the generated values fall outside the range observed during training [1]. In future work, we will explore advanced techniques from the diffusion literature (e.g., [2]) to mitigate this mismatch. Nonetheless, we want to emphasize that our method outperforms the baselines by a wide margin.
>
> - [1] C. Saharia et al., “Photorealistic text-to-image diffusion models with deep language understanding”, NeurIPS, 2022.
> - [2] S. Lin, B. Liu, J. Li, and X. Yang, “Common Diffusion Noise Schedules and Sample Steps are Flawed”, arXiv, 2023.

---

### Official Review · Reviewer_WSFi · 2023-10-31

**Soundness:** 1 poor
**Presentation:** 2 fair
**Contribution:** 2 fair
**Rating:** 5
**Confidence:** 4

**Summary:**

This paper proposes a generative diffusion model (DiffPoseTalk) combined with a style encoder that extracts style embeddings from short reference videos for 3D facial animations driven by speech. The generation process is extended to include the generation of head poses. The model is trained on reconstructed 3DMM parameters from a high-quality, in-the-wild audio-visual dataset.

**Strengths:**

The main contributions are:
- A diffusion-based approach is proposed to jointly generate diverse and stylistic 3D facial motions with head poses from speech.
- A style encoder is developed to extract personalized speaking styles from reference videos, which can be used to guide the motion generation in a classifier-free manner.
- An audio-visual dataset is constructed that includes a range of diverse identities and head poses.

**Weaknesses:**

- In Section 4.1 it is motivated the use of synthesis data as a source for training the proposed architecture. It is also reported the 3D face reconstruction method in Filntisis et al., 2022) based on a 3DMM is used for accurate reconstruction of lip movements. This reconstruction than learning of a set of 3DMM parameters with related lips movements based on these 2D datasets. However, this is not much convincing to me. Lips movements associated to speech are quite subtle and with specificity from each individual. Reconstructing such subtle movements with a 3DMM from a 2D video sequence seems not capable of capturing the reality of movement. In my opinion, authors should convince the readers more about this point which is also fundamental for the proposed approach.
- Table 1 is not fully convincing because the other compared methods were developed for working with different data and so the proposed evaluation could be somewhat biased.
- The supplemental video material is not fully convincing. For some speech the lips movement and the audio track are not well synchronized. The lips movement is also quite synthetic with the lips that are not able to close. The movement is also on the lips only without facial expression shown during face lips animation.
- There is not much discussion on the proposed method. In particular, limitations of the proposed approach are not evidenced.

---
I have read the other reviews, and the authors’ rebuttal. I thank the authors for their answers and revised paper. The rebuttal has clarified some of the unconvincing points in the original submission. Taking this into consideration I changed my score to marginally below the acceptance threshold.
However, I still think the approach is not convincing in the way it produces synchronized lip movements (looking to the supplementary videos synchronization seems rather unrealistic in most of the cases) as well as the comparison is carried out with other methods in the literature.
Overall, the paper is questionably at the bar for ICLR acceptance.

**Questions:**

Q1: authors should make it evident the methods for lips reconstruction can provide results similar to the real one. I understand this is not the focus of the paper but the proposed approach relies on this assumption which is not enough supported and makes evidence in my opinion.
For example authors could compare results obtained with synthetic data and real one using their solution
Q2: an experiments using real data could have been reported (e.g., VOCASET) for a better comparison with other methods.

---

> ### Author Response · Authors · 2023-11-20
> **Reply to Reviewer WSFi (1/2)**
>
> Thank you for the valuable comments.
>
> We understand that your main concern is regarding our use of "3D data reconstructed from 2D datasets" rather than "scanned real 3D data" in our work. We would like to respond to this concern first, and respond to other detailed questions in the next comment.
>
> Firstly, we emphasize that our contribution is to propose a novel and **general** diffusion-based method for generating speech-to-3D animation. Specifically, our method is able to control the speaker's style using any reference style and support the prediction of head movements. All reviewers agree that this application scenario is valuable. We also note that **existing scanned real 3D datasets cannot be used to build algorithms for this new scenario** due to their limited scale and coverage of identities, styles, and head movements. Therefore, there are two possible strategies: one is to collect new scanned 3D real data, and the other is to reconstruct from existing 2D data. The method proposed in this paper belongs to the latter. We are also very much looking forward to someone coming up with methods of the former in the future, and comparing them with ours.
>
> Secondly, **the strategy of using synthetic data from 3D reconstruction is well-established in various SOTA methods**. Due to the limited scale and coverage in existing 3D real datasets, three recent SOTAs [1-3], including two on emotional 3D facial animations generation [2, 3], used reconstructed data from 2D. In addition, [3] also adopted the reconstruction method we used. Additionally, many successful 2D talking head videos methods that use 3DMM as an intermediate face representation, such as [4-7], also utilize reconstructed data to train their speech-to-3D animation generation networks. Thus, **the practice of using reconstructed data is reasonable and well-established in this field**.
>
> We understand your concern about the quality of reconstructed lips. However, this is not a key issue in our method because existing SOTA reconstruction methods can reconstruct reasonable lip shapes most of the time, especially since **we used a method optimized for lip shape reconstruction (Filntisis et al., 2022)**. Quantitative results, qualitative experiments, and user studies in this paper demonstrated that our method achieved satisfactory generation and outperformed baseline methods. Moreover, **our method does not rely on how 3DMM coefficients are obtained**. In fact, the regression-based reconstruction method we currently use achieves satisfactory results. Of course, if an advanced optimization-based reconstruction algorithm is used or an even stronger regression-based reconstruction algorithm that can provide accurate 3DMM coefficients becomes available in the future, our method can readily use it.
>
> Lastly, we would like to point out that our method’s usage of 3DMM coefficients presents several advantages over the methods utilizing scanned 3D meshes.  First, 2D video data is much easier to collect and reconstructed 3DMM data can **cover a broader range**, leading to models with **stronger generalization capabilities**. Second, different from 3D meshes, the lower dimensionality of 3DMM coefficients **reduces computational load and enhances inference speed**, which is crucial for diffusion models. This reduced dimensionality, combined with blendshapes as a prior, **makes the** **learning process easier and improves generalization** [2]. Third, using 3DMM coefficients **facilitates integration with downstream applications** such as driving rig-based gaming avatars or 3DMM-based expression editing [8, 9].
>
> - [1] C. Zhang et al., “3D Talking Face With Personalized Pose Dynamics” , TVCG, 2023.
> - [2] Z. Peng et al., “EmoTalk: Speech-driven emotional disentanglement for 3D face animation”, ICCV, 2023.
> - [3] R. Daněček, K. Chhatre, S. Tripathi, Y. Wen, M. J. Black, and T. Bolkart, “Emotional Speech-Driven Animation with Content-Emotion Disentanglement”, SIGGRAPH Asia Conference, 2023.
> - [4] J. Thies, M. Elgharib, A. Tewari, C. Theobalt, and M. Nießner, “Neural Voice Puppetry: Audio-driven Facial Reenactment”, ECCV, 2020.
> - [5] Z. Zhang, L. Li, Y. Ding, and C. Fan, “Flow-Guided One-Shot Talking Face Generation With a High-Resolution Audio-Visual Dataset”, CVPR, 2021.
> - [6] W. Zhang et al., “SadTalker: Learning Realistic 3D Motion Coefficients for Stylized Audio-Driven Single Image Talking Face Animation”, CVPR, 2023.
> - [7] C. Xu et al., “High-Fidelity Generalized Emotional Talking Face Generation With Multi-Modal Emotion Space Learning”, CVPR, 2023.
> - [8] Z. Geng, C. Cao, and S. Tulyakov, “3D Guided Fine-Grained Face Manipulation”, CVPR, 2019.
> - [9] Z. Sun et al., “Continuously Controllable Facial Expression Editing in Talking Face Videos,” arXiv, 2022.

---

> ### Author Response · Authors · 2023-11-20
> **Reply to Reviewer WSFi (2/2)**
>
> Q1: Fairness in Table 1 for the methods that were designed for working with different data.
>
> A1: We understand your concern: FaceFormer/CodeTalker were designed for animation with fixed styles and no head movements, while our method is designed for 3D face animation that **generalizes to unseen styles along with head movements**, thus the comparison may appear unfair. **Our comparison follows previous work** [2, 3]. In particular, we train these methods with zero-posed mesh data converted from 3DMM coefficients (which matches the original data's topology) and provide one-hot speaker labels **for fair comparison**. Additionally, the integration of generalizable style and head movements are essential components towards more natural and realistic 3D face animation.
>
> Q2: Supplemental video: for some parts, the lip movements and the audio track are not well synchronized, and movement is on the lips only without facial expression shown during face lips animation.
>
> A2: First, we clarify that **our method can automatically generate upper face motions**, such as eye blinks (e.g., in demo video 2:07-2:20), that are weakly-related or unrelated to speech, as stated in “Qualitative Evaluation” in Sec. 4.3. In contrast, the other methods show less upper face motions. Our FDD metrics also surpass other methods by a wide margin, **which indicates that our upper face motions are closer to the ground truth**. This improvement is largely because we adapted the diffusion model for facial animation generation. The probabilistic approach can capture the data distribution of speech-unrelated movements more effectively than regression-based methods. Second, our method **shows significant improvements in lip synchronization, style control, and naturalness over previous SOTA methods**, as demonstrated by our quantitative results, qualitative experiments, and user studies.
>
> Q3: More limitation discussions on the proposed method.
>
> A3: The following discussions on limitations have been added to the revised appendix.
> “Although our method is able to generate high-quality stylistic 3D talking face animation with vivid head poses, there are still some limitations within our framework that could be addressed in follow-up works. Firstly, the computational cost of inference is relatively high due to the sequential nature of the denoising process. To mitigate this, future research can explore more advanced denoisers such as DPM-solver++. Secondly, like existing SOTAs, our approach focuses on animating the face shape while ignoring the inner mouth (including teeth and tongue). Exploring representation and animation of the inner mouth can lead to more realistic results.  Lastly, a promising direction for future research would be to collect real-world 3D talking data that encompasses a broader range of identities and styles, which would further enhance the effectiveness of our approach and contributes to the research community.”
>
> Q4: Evidence that lips reconstruction can provide results similar to the real one.
>
> A4: We agree that existing SOTA reconstruction methods may not provide 3D data as accurately as scanned real data. However, we note that the reconstructed data is already feasible for capturing the dynamics of the lips. As mentioned in the previous comment, the use of reconstructed 3D data from 2D datasets is a **reasonable and well-established practice** [1-7]. Furthermore, the reconstruction method (Filntisis et al., 2022) we adopted was **designed and optimized for accurate lip shape reconstruction**. As demonstrated in their paper and demo video, this method can yield plausible lip reconstructions from 2D data.
>
> Q5: Report an experiment using real data (e.g., VOCASET) for a better comparison.
>
> A5: We have compared our method with FaceFormer and CodeTalker on the VOCASET. We list the quantitative results here. More details about the experiment can be found in Sec. A.6 of the revised appendix, and qualitative results can be found in the updated demo video (5:12-5:31).
>
> | Method     | LVE (mm) $\downarrow$ | FDD ($\times 10^{-5}$m $\downarrow$) |
> | ---------- | ----------- | ----------- |
> | FaceFormer | 4.43    | 13.61    |
> | CodeTalker | 4.55    | 10.34    |
> | Ours       | **4.35** | **8.85** |

---

> ### Author Response · Authors · 2023-11-22
>
> Dear reviewer,
>
> We would be most grateful if you could kindly provide us with your feedback regarding whether our response has effectively addressed your concerns. If there are any further queries or issues, we would be more than happy to answer you.

---

> ### Author Response · Authors · 2023-11-23
>
> Dear Reviewer,
>
> With less than 12 hours remaining until the end of the rebuttal period, we would greatly appreciate any additional comments and suggestions you may have regarding our work. We hope that our responses have addressed your concerns. Thank you sincerely!

---

> ### Author Response · Authors · 2023-11-23
>
> Dear Reviewer,
>
> As the deadline for rebuttal approaches, we hope we have adequately addressed your concerns. We would greatly appreciate it if you could reconsider your evaluation in light of the response and additional materials provided. Thank you for your valuable feedback!

---

### Official Review · Reviewer_xqkb · 2023-11-01

**Soundness:** 3 good
**Presentation:** 3 good
**Contribution:** 3 good
**Rating:** 5
**Confidence:** 4

**Summary:**

The paper presents a method for audio-driven facial animation, where reference style is modeled using a reference video (3D tracking of the video) encoder (and not generic one-hot encoding) and diffusion-based denoising network for final vertex animation. The main contribution of the paper is to introduce the use of diffusion (similar to Tevet '23 et al) for audio-driven talking head generation. While overall no new loss functions (except a simple smoothness loss) are introduced, however, the end-to-end formulation is novel, along with reference style encoding using transformers and contrastive learning and head pose movement. Finally, like previous approaches W2V2 is used for audio feature extraction. The paper also introduces a small dataset of tracked videos with richer set of emotion and style variations. Overall the method outperforms several SOTA methods wrt to quality of articulation and style transfers and provides ablations for their method.

**Strengths:**

- A novel formulation for target style-encoding that's able to extract salient features from target video via tracking and use these for "style-transfer".
- Lack of high-quality data in this space is a big problem, and hence the small dataset release for audio-driven facial animation is a welcome contribution.
- Diffusion-based approach for final audio-driven facial animation. Given that for this part the work build up on Tevet '23 et al, it's a less novel contribution
- The SOTA comparisons and user-studies suggest the proposed solution provides improvement over previous approaches.

**Weaknesses:**

- While the paper claims that they can meaningfully extract "style" from the reference videos, in the examples shown in the video (2:42), they don't show the reference styles so it's hard to understand how faithfully the reference style was matched, further only a a single example for the same is shown.
- In the user-study the authors claim that they have superior scores, but none of the user study samples are shared, so it's hard to understand what the users were really scoring.
- The paper does not discuss or define, what they define as style, and how they measure "style" disentanglement from "content".

**Questions:**

- What metric will you use to measure the disentanglement between style and content? Can you show a couple of example of style or content muted performances (where the original video is assumed to have both)?
- Can you explain the motivation behind the contrastive loss for your problem? It's unclear why contrastive training makes sense intuitively.

---

> ### Author Response · Authors · 2023-11-20
> **Reply to Reviewer xqkb**
>
> Thank you for the valuable comments.
>
> Q1: Contribution novelty compared to Tevet '23 et al.
>
> A1: Although our usage of diffusion is similar to Tevet ’23 et al., we introduced several special designs to make the diffusion model work for speech-driven facial animation. First, different from the transformer encoder structure of the denoising network in Tevet ’23 et al., we followed previous successful non-diffusion-based speech-driven facial animation methods to adopt a transformer encoder-decoder structure, where the Wav2Vec2 encoder provided robust speech features. Second, we introduced the alignment mask from FaceFormer to align speech features and motion features, without which the model was unable to generate synchronized facial motions. Third, while Tevet ’23 et al. has a maximum limit on the generated sequence, we introduced a windowing design and a random truncation strategy to allow arbitrary length generation. Finally, we added style condition and achieved style control with incremental classifier-free guidance.
>
> Q2: Show more examples and the reference styles in video for better understanding.
>
> A2: We have added more examples with both the style reference videos and the generated videos in the updated demo video (2:34-3:53).
>
> Q3: Share user study samples for better understanding.
>
> A3: We have added screenshots and more detailed description of our user study in Sec. A.5 of the revised appendix.
>
> Q4: Discuss or define what is style.
>
> A4: As stated in the first paragraph of Sec. 3.3.1, “Speaking style is a multifaceted attribute that manifests in various aspects such as the size of the mouth opening (Cudeiro et al., 2019), facial expression dynamics — especially in the upper face (Xing et al., 2023) — and head movement patterns (Yi et al., 2022; Zhang et al., 2023a)”. Given the complexity and difficulty in quantitatively describing a speaking style with an exact metric, we adopted several metrics to evaluate different speaking behavior characteristics. As stated in “Quantitative Evaluation” of Sec. 4.3, “FDD evaluates the upper face motions, which are closely related to speaking styles, by comparing the standard deviation of each upper face vertex’s motion over time between the prediction and ground truth.” The beat alignment score (BA) was used to “assess head motion”, and the “mouth opening difference (MOD), which measures the average difference in the size of the mouth opening between the prediction and ground truth”, also reflects the speaking style difference in mouth opening.
>
> Q5: Metric to measure the disentanglement between style and content, and show more examples of style or content muted performances
>
> A5: Similar to the challenges in quantitatively evaluating the disentanglement of emotion and content in EmoTalk (Peng et al., 2023), quantitatively measuring the disentanglement of style and content presents similar difficulties. However, it is possible to evaluate this qualitatively. To illustrate, we did a qualitative experiment: we randomly selected 10 speaking styles and 20 audio clips to generate 10*20 animations. The speaking styles from these animations were then extracted using our style encoder. We employed t-SNE for visualizing these extracted speaking styles. The results demonstrated that animations with identical reference styles cluster together, regardless of content differences. For more details, please refer to Sec. A.4 in the revised appendix . In addition, we also show examples of style or content muted performances in the demo video (4:28-4:46).
>
> Q6: Motivation behind the contrastive loss.
>
> A6: As stated in the first paragraph of Sec. 3.3.1, “Given the complexity and difficulty in quantitatively describing speaking styles, we opt for an implicit learning approach through contrastive learning. We operate under the assumption that the short-term speaking styles of the same person at two proximate times should be similar.” Note that many works were also built on similar assumptions: Yi et al. (2022) assumes the speaking style within the same video is similar. FaceFormer and CodeTalker assume the speaking style from the same person is similar.

---

> ### Author Response · Authors · 2023-11-22
>
> Dear reviewer,
>
> We would be most grateful if you could kindly provide us with your feedback regarding whether our response has effectively addressed your concerns. If there are any further queries or issues, we would be more than happy to answer you.

---

> ### Author Response · Authors · 2023-11-23
>
> Dear Reviewer,
>
> With less than 12 hours remaining until the end of the rebuttal period, we would greatly appreciate any additional comments and suggestions you may have regarding our work. We hope that our responses have addressed your concerns. Thank you sincerely!

---

> ### Author Response · Authors · 2023-11-23
>
> Dear Reviewer,
>
> As the deadline for rebuttal approaches, we hope we have adequately addressed your concerns. We would greatly appreciate it if you could reconsider your evaluation in light of the response and additional materials provided. Thank you for your valuable feedback!

---

### Author Response · Authors · 2023-11-20
**General Response**

We thank all the reviewers for valuable comments that help us improve this paper. We greatly appreciate reviewers pointing out that DiffPoseTalk’s **“end-to-end formulation” and “target style-encoding” are novel** (xqkb); the diffusion-based method is **“technically sound”** and the style encoder **“offers customizability to the generated face motions and improves the utility of the proposed method”** (rH7c); our proposed dataset is **“a welcome contribution”** to solving the “lack of high-quality data in this space” (xqkb); our method is **“evaluated well”** (3BSS) and **“outperforms several SOTA methods wrt to quality of articulation and style transfers”** (xqkb), and our **“manuscript is clear and easy to follow”** (3BSS).

We have revised the paper (marked in green) and demo video to address review comments. Here is a summary of the changes:

- Main text
  - Fix a typo in Table 1.
- Appendix
  - Add screenshots of the user study system and samples (xqkb)
  - Add visualization of the style and content disentanglement (xqkb)
  - Add detailed limitations (WSFi, 3BSS) and ethics discussions (3BSS)
  - Add quantitative evaluations on VOCASET (WSFi)
- Demo video
  - Add more examples with both style reference videos and generated videos [2:34-3:53] (xqkb)
  - Add examples of style or content muted performances [4:28-4:46] (xqkb)
  - Add qualitative comparisons on VOCASET [5:12-5:31] (WSFi)
  - Add examples of one-to-many generation [3:55-4:26]

Once again, we would like to express our sincerest gratitude to all the reviewers. Please find detailed responses in the individual comments.

---

### Meta-Review · Area_Chair_iD3E · 2023-12-10

**Metareview:**

The paper introduces 'DiffPoseTalk', a novel generative diffusion model combined with a style encoder designed for creating 3D facial animations driven by speech. This work has been thoroughly reviewed by four experts in the field. The reviewers have recognized and appreciated the paper's novel formulation for target style encoding. This method's ability to extract salient features from target videos via tracking and utilize them for style-transfer is seen as a contribution.

However, despite these positive aspects, there remain pivotal concerns that preclude a recommendation for acceptance at this stage. A primary issue raised by the reviewers pertains to the qualitative results of the synchronized lip movements. The current demonstration of these movements does not sufficiently convey realism, which is crucial for establishing the method's superiority over existing techniques. Improving the realism and effectiveness of these animations is essential for demonstrating the full potential and impact of the proposed method.

While the paper demonstrates clear merit, the decision, considering the reviewers' feedback, leans towards not recommending acceptance in its present form. We encourage the authors to take the reviewers' feedback into consideration, particularly focusing on enhancing the realism of the lip synchronization aspects of the animations. A revised manuscript that addresses these key issues, especially improving the qualitative aspects of the lip movements, could significantly strengthen its contribution to the field of 3D facial animation.

**Justification For Why Not Higher Score:**

The qualitative results of synchronized lip movements are not realistic enough to demonstrate the superiority of the proposed method.

**Justification For Why Not Lower Score:**

N/A

---

### Decision · Program_Chairs · 2024-01-16

Reject